# Fluorescence and Nonlinear Optical Response of Graphene Quantum Dots Produced by Pulsed Laser Irradiation in Toluene

**DOI:** 10.3390/molecules27227988

**Published:** 2022-11-17

**Authors:** Parvathy Nancy, Nithin Joy, Sivakumaran Valluvadasan, Reji Philip, Sabu Thomas, Rodolphe Antoine, Nandakumar Kalarikkal

**Affiliations:** 1School of Pure and Applied Physics, Mahatma Gandhi University, Kottayam 686560, India; 2School of Energy Materials, Mahatma Gandhi University, Kottayam 686560, India; 3Light & Matter Physics Group, Raman Research Institute, Bengaluru 560080, India; 4Accelerator Division, Institute of Plasma Research, Near Indira Bridge, Ahmedabad 382428, India; 5School of Nanoscience and Nanotechnology, Mahatma Gandhi University, Kottayam 686560, India; 6Institut Lumière Matière UMR 5306, Univ Lyon, Université Claude Bernard Lyon 1, CNRS, F-69100 Villeurbanne, France

**Keywords:** pulsed laser irradiation, graphene quantum dots, fluorescence, optical limiting

## Abstract

Graphene quantum dots (GQDs), the zero dimensional (0D) single nanostructures, have many exciting technological applications in diversified fields such as sensors, light emitting devices, bio imaging probes, solar cells, etc. They are emerging as a functional tool to modulate light by means of molecular engineering due to its merits, including relatively low extend of loss, large outstretch of spatial confinement and control via doping, size and shape. In this article, we present a one pot, facile and ecofriendly synthesis approach for fabricating GQDs via pulsed laser irradiation of an organic solvent (toluene) without any catalyst. It is a promising synthesis choice to prepare GQDs due to its fast production, lack of byproducts and further purification, as well as the control over the product by accurate tuning of laser parameters. In this work, the second (532 nm) and third harmonic (355 nm) wavelengths of a pulsed nanosecond Nd:YAG laser have been employed for the synthesis. It has been found that the obtained GQDs display fluorescence and is expected to have potential applications in optoelectronics and light-harvesting devices. In addition, nonlinear optical absorption of the prepared GQDs was measured using the open aperture z-scan technique (in the nanosecond regime). These GQDs exhibit excellent optical limiting properties, especially those synthesized at 532 nm wavelength.

## 1. Introduction

Graphene quantum dots (GQDs) are a distinguished zero-dimensional group of graphitic nanomaterials having various topologies. They are minute splinters of carbonaceous materials with particle size below 10 nm and an sp^2^-carbon structure core, revealing exciton confinement and quantum size effect [1,2]. Generally, the fabrication of broadband modulators, solar cells, photodetectors, etc., were based on the materials having broadband absorption (BBA) and emission that covers the UV-Visible, near-IR regions of the electromagnetic spectrum [3,4,5,6,7,8]. The materials with BBA also exhibit non-linearity and hence they can have potential applications in the fields of high harmonic generation, optical parametric oscillation, multiphoton imaging and Kerr effect [9,10,11,12]. Hence, inventing novel materials with both optical nonlinearity and BBA characteristics are of great interest nowadays. Quantum dots belong to the groups IV to VI, namely, CdS, CdSe, PbSe, ZnSe, HgTe, etc., have already been applied in variety of applications, such as bio imaging, light emitting diodes and solar cells, due to their tunable absorption and desired optical nonlinear characteristics [13,14,15,16,17]. Materials generated from high bandwidth semiconductors, such as GaN, ZnSe, ZnS and AlN, exhibit UV optical activity, and materials generated from rare earth doped GaN and CdS possess near IR activity [18,19]. Even though the size tuning of quantum dots can alter the active optical range, it cannot give the entire range completely, i.e., simultaneous UV-Vis and IR activity. To achieve this, GQDs and modified GQDs are promising candidates for such active optical activity [20,21,22]. Hence, GQDs are a better substitute for these toxic groups IV-VI quantum dots due to their biocompatibility, superior photo stability and cost-effective preparation methods.

Regardless of the synthetic route, GQDs display photoluminescence (PL) features. The PL emission is heavily dependent on the morphology and micro structure instigated during the synthesis. Despite the enormous effort paid to this field of research, the exact origin of photoluminescence (PL) and the fate of the excitation energy are still under debate. In the past couple of years, GQDs were subjected to extensive study and found that the photoluminescence (PL) emission may be aroused from electron-hole recombination, doping, quantum size effect, surface defects in the functional groups of GQDs, etc. [23]. In addition to PL, GQDs may exhibit interesting nonlinear optical properties. Recently, lots of theoretical and experimental research has been conducted on the BBA of GQDs of different sizes, shapes and functionalities, and great progress has been registered in elucidating the shape and size dependent nonlinear activity of GQDs [24,25,26,27,28,29]. In particular Ma et al., using z-scan techniques with femtosecond laser at 532 nm, investigated the third-order nonlinear optical responses of carbon dots and found that both samples exhibited positive nonlinear refractivity and negative nonlinear absorption coefficient [30].

Concerning the routes to produce GQDs, various methods, such as solvothermal, hydrothermal, electrochemical synthesis, ultrasonic, pyrolysis of organic compounds etc., have been incorporated to synthesize GQDs with control in microstructure and morphology [31,32,33,34,35,36]. However, many of these methods were complicated, time consuming and required pre-treatment and heavy purification processes. Later, the researchers focused on tuning the optical properties, chemical and surface characters of the GQDs through doping with heteroatoms, namely, Si, N, B and S [36,37,38,39]. Almost all these methods required costly precursors and long reaction time. Patil et al. introduced pulsed laser ablation in liquid (PLAL) as a novel, comparatively easier and single step to synthesize nanoparticles in 1987 [40,41]. In PLAL, a solid target is submerged in a liquid medium, and ablated with a beam of high energy Laser. During this process, the interaction between intense (~10^6^ to 10^14^ W/cm^2^) and short (~10^−13^ to 10^−8^ s) pulses of laser and the interface between the surface of the target material and liquid would generate bubbles with high temperature and pressure. This results in vaporization and the subsequent emission of surface materials brings nucleation and nanoparticle generation [42,43]. A couple of studies were reported on pulsed laser ablation of solid carbon targets in liquid media to produce carbon dots and GQDs. Furthermore, carbon nanoparticles generated by PLAL show solvent dependent optical properties since the surface can be functionalized or modified by the reaction with the solvent [44,45,46,47]. Pegylated GQDs were generated in situ through PLAL of a graphite target in polyethylene glycol solution as in-vivo fluorescent image agent [48]. It is also reported that N-doped GQDs can be fabricated through PLAL synthesis by using aqueous solution of graphene oxide as carbon source and diethylene triamine as N source [49]. These reports revealed the possibility of surface passivation and doping with heteroatoms, which can augment the quality of GQDs. Recently, pulsed laser assisted sythethis has been demonstrated as a clean and efficient one-step method to fabricate GQDs and carbon nanostructures from a bottom-up approach using precursors such as toluene [50], chlorobenzene and o-dichlorobenzene [51].

Using a similar bottom-up approach, the motivation of this work is to generate GQDs in toluene, by pulsed laser irradiation, without using many reagents and catalysts. A veriety of GQDs are produced by using both the second (532 nm) and third harmonic (355 nm) wavelengths of an Nd:YAG laser. The GQDs exhibit unique optical limiting properties, especially those synthesized at 532 nm wavelength. This study highlights the impact given by nanosecond laser radiation upon the size controlled of synthesis of GQDs from in an organic liquid with exceptionally small reaction time. Apart from the previous report by Yu et al. [50], here we explored the GQDs synthesis with second (532 nm) and third harmonic (355 nm) wavelengths of Nd:YAG laser other than the fundamental wavelength (1064 nm) without using any reaction protective gases such as Argon. The whole experiment was conducted in room temperature and atmospheric pressure. We found that, at fluence, F ≥ 3.1 Jcm^−2^, the ablation mechanism is apparently explosive boiling and does not lead to QDs formation and we fabricated less than 10 nm GQDs at lowest laser fluences in these particular experimental conditions, apart from the study reported by Yu et al. [50]. Even though the direct photodissociation of toluene molecules to carbon dots under pulsed laser irradiation in the absence of surfactants or catalysts was also reported by Zhu et al. [52], we had generated GQDs in nanosecond regime and established the size control by varying the laser wavelength and established the effect of the same to its NLO properties for the first time.

## 2. Experimental Details

### 2.1. Synthesis

An Nd:YAG Laser beam (Litron LPY 674G-10, repetition rate-10 Hz, pulse width-8 ns) was focused into a glass cuvette containing 30 mL of toluene and ablated that carbon source for 120 s, corresponding to different laser energy densities (starting from 1 Jcm^−2^ to 3 Jcm^−2^), which immediately results in the generation of graphene quantum dots (see Figure 1). The experiments have been performed at second (532 nm) and third (355 nm) harmonic laser wavelengths. Here, the bottom-up synthesis approach for generating highly pure colloidal GQDs has been employed. Laser irradiation creates high non-equilibrium conditions within liquids with extremely high pressure and temperature where anomalous processes and the fast growth of the fragmented nano-sized species can occur. A change in color of the solution to transparent yellow after the irradiation has been observed (Figure 1), which infers the formation of GQDs. Further, the samples were centrifuged at 6000 rpm for 20 min and then the GQDs will remain in the supernatant liquid.

### 2.2. Characterization

The FTIR spectra of the samples were recorded using ATR-FTIR (Shimadzu, IR Prestige 21) spectrometer in the range of 4000–450 cm^−1^. The Raman spectra were recorded with Confocal Raman Microscope with AFM WITec Germany, Alpha 300RA. TEM measurements were performed on JEOL JEM 2100 electron microscope at operating voltage of 200 kV. Fluorescence spectra were recorded on a fluorescence spectrophotometer (HORIBA Fluoromax-4 spectrofluorometer). Prompt decay is the instrumental response function, the response of the instrument to a zero-lifetime sample. This curve is typically collected using a dilute scattering solution such as colloidal silica (Ludox) and no emission filter. This decay represents the shortest time profile that can be measured by the instrument. The width of the IRF is due to both the characteristics of the detector and the timing electronics. Shimadzu (model 2450) UV-Vis spectrophotometer was used for the absorption measurements and the spectra were typically measured in the range of 85–800 nm. X-ray photoelectron spectroscopy (XPS) was carried out using Kratol Analytical, Axis ultra 1486 eV spectrometer with monochrome Al Kα as the excitation source. The nonlinear optical properties (NLO) of the GQDs have been examined by the z-scan technique as described in reference [53].

## 3. Results and Discussion

The GQDs via pulsed laser irradiation of toluene have been prepared within a short span of time. When the laser beam impinges on the surface of the toluene, the electrons are under the influence of the electric field of the laser pulses and are accelerated to produce plasma. This laser produced plasma enhances the breakdown of toluene to form GQDs. The experiments have been performed at second (532 nm) and third (355 nm) harmonic laser wavelengths with laser fluence ranging from 1 to 3 Jcm^−2^. At large fluences (F ≥ 3.1 Jcm^−2^), the irradiation mechanism becomes explosive and that does not lead to GQD formation. The preparation requires utilization of very small reaction time (2 min) and less laser energy compared to the previous reported literatures [6,7]. The physical, chemical and morphological characteristics were studied by, FTIR and Raman spectroscopy, X-ray photoelectron spectroscopy and Transmission Electron Microscopy. The optical characteristics were studied by UV-Visible spectroscopy and photoluminescence spectroscopy. The nonlinear optical response of the GQDs has been examined by z-scan technique.

### 3.1. Morphological Analysis

The structure and morphology of the as-prepared GQDs were investigated by transmission electron microscopy (TEM). Transmission Electron Microscopy (TEM) images were recorded using JEOL JEM 2100 electron microscope. Figure 2 illustrates the TEM images of GQDs, in which GQD@355 nm displays particle size in the range of 3 ± 2 nm, and GQD@532 nm in the range 5 ± 3 nm. It has been noted that there is a slight increase in the particles size of GQDs with the long wavelength laser ablation. Almost all the GQDs show uniformity in size and shape. The particle size distribution curve is included in the inset. The HRTEM images reveal clear details about the inter planar distance of 0.212 nm for the GQD which is tie in with the (100) plane of graphene. The selected area electron diffraction (SAED) of the as-prepared GQDs exhibited diffraction rings due to the presence of nanocrystals randomly oriented and most of them are associated with the graphitic structure. The TEM analysis had demonstrated the presence of a graphitic core, unlike an amorphous core in ordinary carbon dots.

### 3.2. XPS, FT-IR and Raman Analyses

XPS measurements were carried out to probe the composition of GQDs. As seen in Figure 3a, the XPS show a dominant graphitic C1s peak at 284.8 eV and O1s peak at 532 eV for GQDs. The measured spectrum of C1s is nicely de-convoluted into four surface components, which correspond to sp2 C (C=C, C–C) at a binding energy of 284.6 eV, sp3 C (C–OH) at 285.9 eV, C–O–C at 287.3 eV and C=O at 289.6 eV [54]. These results suggest that functionalization of the graphene sheet with oxygen containing functional groups took place during the course of the reaction. The presence of sp3 C peak in the XPS spectra suggests that planarity in GQD structure has reduced severely, and defects appeared in the prepared GQDs. The XPS spectrum of O1s is resolved into peaks at 532.7 eV corresponding to C=O and these results suggest that the prepared GQDs are rich in oxygen containing functional groups.

The Raman spectra of GQDs are shown in Figure 3d, where peaks around 1329 cm^−1^ and 1582 cm^−1^ represent the D-band and G-band, respectively. The D and G peaks appeared in the spectrum attributed to the aromatic domains of graphene quantum dots. The peak at 1329 cm^−1^ is associated to the disordered band due to structural defects, dangling and edge sp3 carbon bonds that break the symmetry and the peak at 1582 cm^−1^ correspond to the in plane stretching vibration of sp^2^ carbon atoms. The intensity of the G band is related to the amount of sp^2^ domains whereas the D band to the sp3 domains. The intensity ratio between the D and G bands (I_D_/I_G_) can be employed to examine the sp^3^ structure of the GQDs. There is no appreciable difference in the I_D_/I_G_ ratio (0.94) for both GQD@532 nm and GQD@355 nm. This implies that the synthesized GQDs have more disordered structure and this could be due to the presence of more oxygen-containing functional groups in the prepared GQDs. Consequently, the intensity of Raman scattering was determined by the concentration of particles confirming that the scattered Raman intensity has increased in the case of GQD@532 nm. The functional groups of GQDs were identified by FTIR spectroscopy by analyzing their vibrational spectra. Existence of C=O, O–H and C–O groups in GQDs are substantiated by FTIR spectroscopy. Figure 3e shows the FTIR spectra of GQDs synthesized at two different laser wavelengths. The FTIR spectra of GQDs (in solution form) show highly resolved peaks corresponding to stretching frequency of O–H, C=O and C–O groups. The bands at 2981 cm^−1^ and 1375 cm^−1^ are attributed to the stretching and bending vibrations of C–H groups, respectively. The broad band at 3423 cm^−1^ is assigned to the stretching frequency of O–H groups. The bands at 2981 cm^−1^ and 1375 cm^−1^, respectively, are attributed to the stretching and bending vibrations of C–H groups. The bands around 1728 cm^−1^ and 1225 cm^−1^ are due to the stretching of C=O and C–O groups, respectively. The peak at 1625 cm^−1^ indicates the existence of aromatic C=C groups. It can be seen that the C-H bending and stretching occurs at 1468 cm^−1^ and 2930 cm^−1^ respectively and the C=C bonds shows the presence of aromatic carbon which is situated at around 1735 cm^−1^ [55]. Of note, after being laser irradiated, the FTIR spectra of the GQDs suspension showed the same characteristics as pure toluene (compare spectra in Figure 1e,f), confirming that the solvent after laser irradiation was still toluene.

### 3.3. UV-Visible Absorption Spectroscopy

To obtain the optical properties, UV-Vis analysis of GQDs were carried out. Figure 4a) shows the UV Visible absorption spectra of GQD@532 nm and GQD@355 nm synthesized at 2.6 Jcm^−2^ laser fluence. The absorption spectra of both the prepared samples are almost identical and at 287 nm with a small shoulder peak at around 340 nm that corresponds to the n-π* transitions of the carbon oxygen bond (C=O), which is most obviously seen in graphene [56]. Here, it can also be seen that the absorption intensity increases with increase in the laser wavelength and fluence which indicates the increase in particle size and concentration at higher wavelength ablated species (Figure 4b). Since the absorption cross section is proportional to square diameter of absorbing particles, much smaller absorbance occurred in the case of 355 nm indicating smaller density of absorbing particles. Irradiation with a 355 nm wavelength resulted in a surprisingly small production rate compared to longer wavelengths, such as 532 nm, due to absorption of the laser radiation on a short path.

### 3.4. Photoluminescence Spectra

The photoluminescence properties of the samples are shown in the Figure 5a. The PL emission of GQDs synthesized at 2.6 Jcm^−2^ at second and third harmonics of laser wavelengths and the PL emission is observed at 406 nm at an excitation wavelength 360 nm. From this it could be inferred that the GQDs exhibits blue fluorescence, which is the contribution of emission from LUMO to HUMO [35]. It has also been observed that there is a small luminescence quenching takes place with a little reduction in intensity in the case of GQD@355 nm. Moreover, the PL intensity enhances with the increase in laser wavelength and the strong photoluminescence activity of the prepared GQDs in the ultraviolet-visible region can be explored in various sensing applications.

The fluorescence lifetime of the GQDs (Figure 5b) was investigated using time-correlated single photon counting (TCSPC) equipped with the HORIBA Fluoromax-4 spectrofluorometer. The measurement of the fluorescence lifetime is one of the absolute methods to distinguish between static and dynamic fluorescence quenching. The fluorescence decay curves of the colloidal solution of GQDs were fitted to multi-exponential functions. Here, GOD requires three time constants for satisfactory fits and they are T1 = 3.135083 × 10^−9^ s, T2 = 1.170301 × 10^−9^ s, T3 = 1.05686 × 10^−8^ s respectively. The mean life time value can be calculated using Equation (1) [57].
T_m_ = a_1_T_1_ + a_2_T_2_ + a_3_T_3_(1)
where, a1, a2, a3, T1, T2 and T3 are the relative amplitudes and decay times of the multi exponential components of the fluorescence decay. The mean life time of the prepared GQD obtained from the time resolved measurement is 3.7 ns and the quenching effect was identified as static fluorescence quenching. The prepared GQDs display very low fluorescence quantum yields (estimated to ~10^−4^ to 10^−3^ using the reference method and rhodamine B as reference dye), mainly due to a very strong absorbance in the UV range and absence of a comparable level of absorbance at the wavelengths of maxima of the excitation spectra.

Commission International d’Eclairage (CIE) 1931 system has been used to evaluate the color luminescence emission intensities and to identify the dominant emission wavelength of the fabricated GQDs for evaluating the material performance which is shown in Figure 5c. CIE chromaticity studies can be useful in this context to give more idea about the possible application in LED lighting devices. The color of the phosphor can be frequently expressed by (CIE) coordinates. To reflect the true color of GQD luminescence, CIE 1931 (x, y) co-ordinates are calculated [58,59]. The CIE coordinates calculated are (0.175, 0.178). Hence, the prepared GQDs while excited with commercially available blue LED’s can be used as potential candidates for blue emitting light sources.

### 3.5. Nonlinear Optical Studies

The nonlinear optical transmission measurements were carried out using the open aperture z-scan technique, which is actually a measurement of the optical transmission of the material as a function of input light intensity. The z-scan set up is comprised of a pulsed laser beam having known energy, a converging lens to focus the laser beam, and laser energy detectors. The propagation direction of the laser beam is assumed to be in the z-axis, such that the fluence is maximum at the focal point z = 0 which decreases towards either direction on the z-axis. The sample is moved along the z-axis, towards or away from the focal point to vary the light fluence falling on it. The position versus transmission curve (z-scan curve) can be drawn by measuring the transmitted energy for different sample positions using a detector. Optical nonlinearity coefficients can then be calculated numerically by fitting the measured z-scan curve to standard nonlinear optical transmission equations [60,61,62].

The laser sources used in the current setup is a Q-switched Nd:YAG laser (Minilite, continuum) emitting at the second harmonic wavelength of 532 nm. The Gaussian beam is focused through a plano-convex lens of focal length 10.75 cm. The laser pulse widths (FWHM) are approximately 5 ns. The samples are taken in a 1 mm cuvette. The cuvette is mounted on a stepper motor controlled linear translational stage and is translated along the z-axis through the focal region. The sample sees different laser fluencies at each z positions and the corresponding transmittance is measured using a pyroelectric energy probe (Rj7620, Laser probe Inc., Utica, NY, USA) placed behind the sample. In all measurements, the lasers were running in a single shot mode by externally triggering the laser output. Figure 6 shows the open aperture z-scan curves and the subsequent fluence dependent nonlinear transmissions acquired for GQDs for 532 nm and 355 nm, 5 ns laser excitation at input laser pulse energy of 25 µJ. The optical limiting measurements are quantified using the limiting threshold value, which is defined as the input fluence at which the sample’s normalized transmittance drops to 50% of its linear transmittance. Materials having lower limiting thresholds exhibit maximum optical limiting performance. The limiting threshold value of GQDs is better than that of standard materials such as C60 in toluene and carbon black in water [63]. Mostly, the optical limiting property of a material can obtain contributions from phenomena such as excited state absorption, (ESA) including free-carrier absorption, excited singlet or triplet absorption, etc., and two or three-photon absorption, which strongly depend on the material, excitation pulse width and wavelength. It has been shown from the previous reports that ESA is much stronger for nanosecond pulse excitations in graphene-based materials [64].

The intensity dependent effective nonlinear absorption co-efficient α (I) is given by the equation [65]
α(I) = αo/(1 + ((I)/(I_sat_))) + β_eff_ I(2)
where I and I_sat_ represent the laser input and saturation intensities, respectively, while α0 and β_eff_ represent the linear absorption and effective two-photon absorption coefficients respectively.

Optical limiters are devices which are active at high intensity as opaque materials, whereas at low intensity, they act like transparent materials. Therefore, these devices are indispensable in optical sensors and eye protective devices. Figure 6 depicts input fluence plotted against normalized transmittance using open aperture z-scan data. GQDs exhibit optical limiting nature as the normalized transmittance decreased with the input fluence. Results reveal that the optical limiting efficiency of GQD@532 nm sample was significantly higher than that of GQD@355 nm, which may be explained by the increase in particle size and concentration at higher wavelength ablated species.

In this work, two photon absorption (TPA) became the effective nonlinear absorption mechanism when using the ns laser pulses. TPA is an instantaneous nonlinearity that involves the simultaneous absorption of two laser photons with assistance of the intermediate state. TPA behavior was strongly affected by the pulse duration and the field intensity. The values of the TPA cross-section increased with the broadening of the incident laser pulse and increasing intensity. Therefore, the TPA effect became much weaker when irradiated with a short laser pulse that is faster than the intersystem transition. Under the laser irradiation with low energy, the available ground state carriers were depleted, while the excited states became almost occupied, and, thus, the Pauli Exclusion Principle came into effect [66]. Moreover, it has exhibited the reverse saturable absorption (RSA) nature of GQDs. The optical limiting threshold is found to be 0.45 Jcm^−2^ for the laser synthesized GQDs@532 nm. The obtained nonlinear parameters for nanosecond laser pulse excitations are given in Table 1.

## 4. Conclusions

The optical properties of GQDs synthesized using nanosecond laser pulses with a carbon source followed by a fast and one-step process have been studied critically. Absorption and photoluminescence spectrumgives direct evidence of the formation of GQDs. Imaging using TEM is sufficient to prove the existence of spherical GQDs having homogenous size less than 10 nm. The as-prepared GQDs exhibit outstanding photoluminescence and chemical stability and the preparation requires utilization of very small reaction time (2 m) and less laser energy. The fluorescent nature of GQDs is trustworthy for biological and optoelectronic applications. It may also be noted that by controlling the laser parameters such as laser fluence, laser wavelength and repetition rate, one can fine-tune different properties such as concentration, particle size, etc., as per the application requirements. Moreover, the third-order-like NLO properties of these GQDs is investigated using open aperture z-scan technique which revealed the RSA nature of GQD so that it can be used as excellent optical limiter especially GQDs synthesized at 532 nm wavelength. These GQDs can be used as an efficient two-photon probe for bio-imaging applications due to their strong TPA. Conclusively, the laser ablation technique, thus, is a successful method for preparing GQDs within a very short span of time, while the GQDs synthesized at longer wavelengths and laser fluences have improved optical properties.

## Figures and Tables

**Figure 1 molecules-27-07988-f001:**
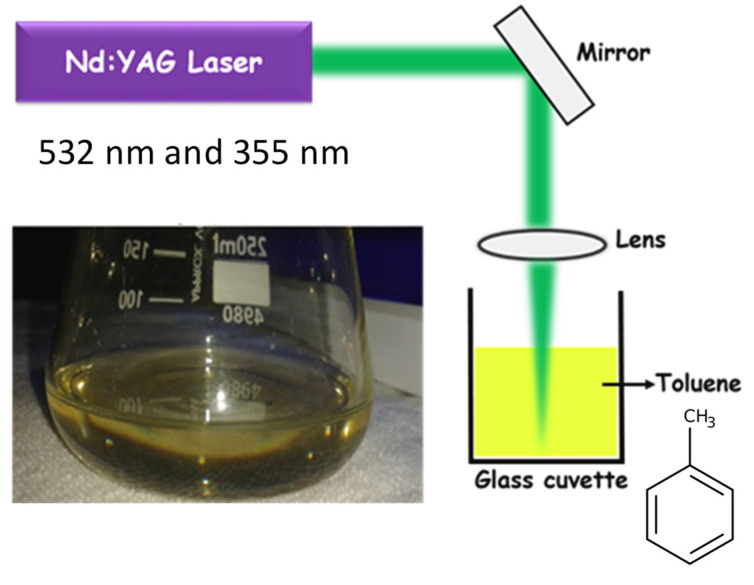
Experimental layout of production of GQD via laser irradiation in toluene and the yellowish GQD solution just after irradiataion.

**Figure 2 molecules-27-07988-f002:**
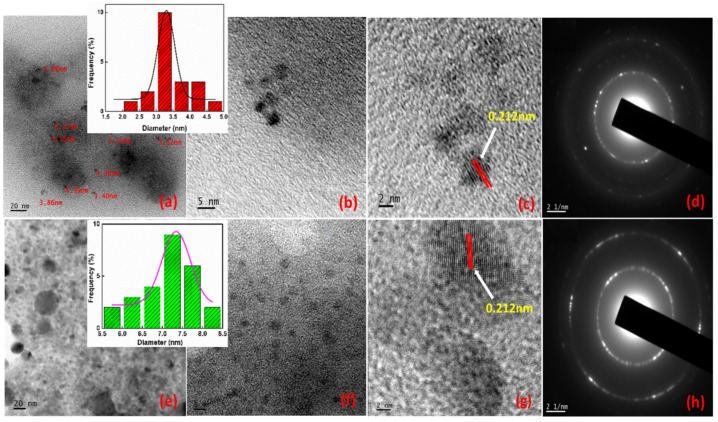
TEM and HRTEM images of (**a**–**d**) GQD@355 nm and (**e**–**h**) GQD@532 nm.

**Figure 3 molecules-27-07988-f003:**
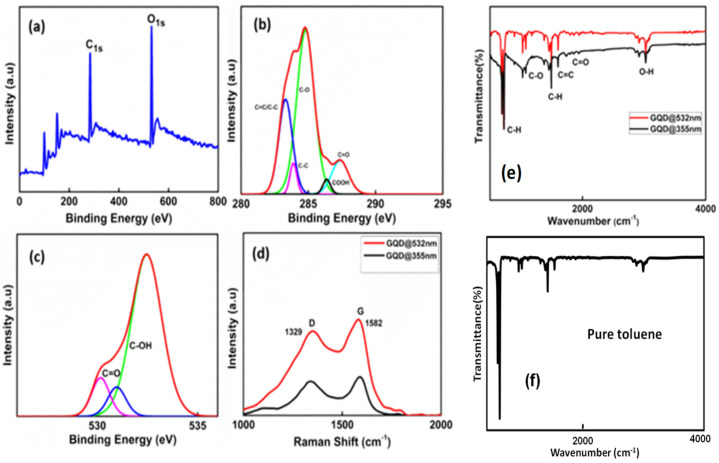
X-ray photoelectron spectroscopy (XPS) and Raman spectra of the laser synthesized GQDs. (**a**) Wide scan spectra. (**b**) C1s spectrum. (**c**) O1s spectrum. (**d**) Raman spectra. (**e**,**f**) FT-IR spectra.

**Figure 4 molecules-27-07988-f004:**
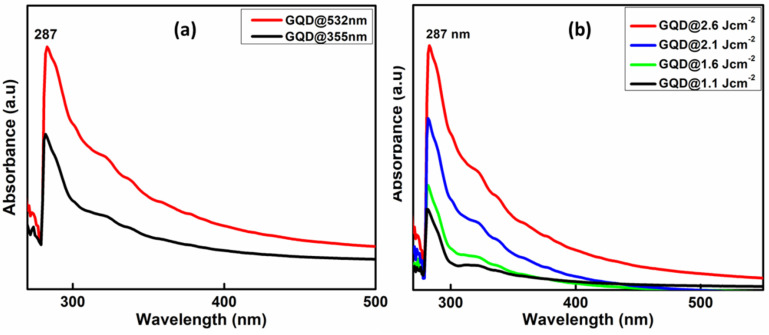
(**a**) UV-Visible absorption spectra of GQDs synthesized at different laser wavelengths and (**b**) at different laser fluences.

**Figure 5 molecules-27-07988-f005:**
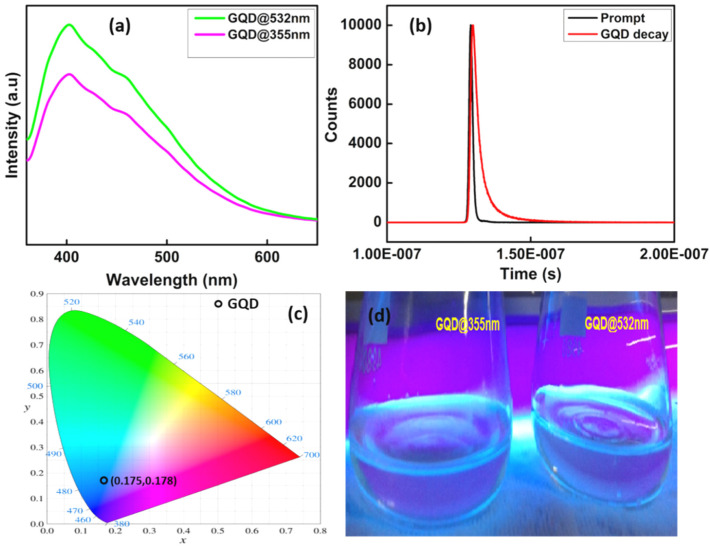
(**a**) Photoluminescence spectra of GQDs at an excitation wavelength 380 nm. (**b**) Fluorescence decay curve. (**c**) CIE chromaticity color coordinate diagram fluorescence to naked eye shown by the samples when subjected to UV radiation (**d**).

**Figure 6 molecules-27-07988-f006:**
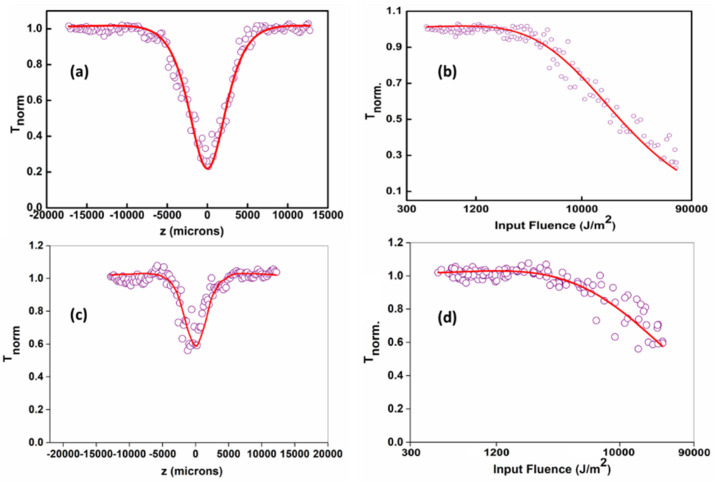
(**a**,**b**) Open aperture z-scan of GQDs@532 nm, 5 ns laser pulse excitation at laser pulse energy of 25 µJ. (**c**,**d**) Open aperture z-scan of GQDs@355 nm, 5 ns laser pulse excitation at laser pulse energy of 25 µJ. Open circles: data points, solid lines: numerical fits to the data points.

**Table 1 molecules-27-07988-t001:** Nonlinear absorption coefficient (β_eff_) and saturation intensity (I_Sat_) for ns excitations, obtained by numerical fits to the data points.

Sample	Energy (µJ)	β_eff_ (× 10^−10^ mW^−1^)	I_sat_ (× 10^10^ Wm^−2^)	Optical Limiting Threshold (Jcm^−2^)
GQDs@532 nm	25	4.3	57.7	0.45
GQD@355 nm	25	0.48	139.9	6.4

## Data Availability

Not applicable.

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
