# Peer review of "Fluorescence and Nonlinear Optical Response of Graphene Quantum Dots Produced by Pulsed Laser Irradiation in Toluene"

_molecules, 2022, doi:10.3390/molecules27227988_

Round 1
Reviewer 1 Report
I recommend acceptance with Minor Revision.
The authors present a one pot, facile and ecofriendly synthesis approach for fabricating GQDs via pulsed laser ablation of toluene without any catalyst, which displays fluorescence and low toxicity. This work is interesting for publication in Molecules after careful revision. The following points need to be well addressed prior to publication.
1. There’s no PL quantum yields data for the GQDs, it’s better to provide them.
2. In Figure 2 a and e, the particle size distribution in the illustration is too vague and needs to be redrawn.
3. In Figure 4, how does the author determine the concentration of the samples? Are these samples prepared based on the same preparation and purification process.
4. In the reference, the format is too messy and needs to be unified. Such as, the order of the author's last name, the full name or abbreviation of the journal, the format of issue and volume.
Author Response
- There’s no PL quantum yields data for the GQDs, it’s better to provide them.
Our response: The prepared GQDs display very low fluorescence quantum yields (estimated to ~10^-4 to 10^-3 using the reference method and rhodamine B as reference dye), mainly due to a very strong absorbance in the UV range and absence of a comparable level of absorbance at the wavelengths of maxima of the excitation spectra.
- In Figure 2a and e, the particle size distribution in the illustration is too vague and needs to be redrawn.
Our response: It is rectified in the revised manuscript.
- In Figure 4, how does the author determine the concentration of the samples? Are these samples prepared based on the same preparation and purification process?
Our response: The concentration can be determined with the help of Beer-Lambert Law. The samples are synthesized at second (532 nm) and third (355 nm) harmonic laser wavelengths at laser fluence 2.6 Jcm-2. The purification process was the same for both the samples as described in the section 2.1.
- In the reference, the format is too messy and needs to be unified. Such as, the order of the author's last name, the full name or abbreviation of the journal, the format of issue and volume.
Our response: The authors are greatly thankful to this suggestion. It is corrected in the revised version.
Reviewer 2 Report
Authors report the synthesis of graphene QDs using pulsed laser ablation of toluene (precursor) at two different laser wavelengths. Graphene QDs were adequately characterized using UV-Vis, TEM, XPS, FTIR, and Raman spectroscopy.
My main concern is the novelty of synthesis employed in this study. Reference 66 (Yu et al., Preparation of carbon dots by non-focusing pulsed laser irradiation in toluene, Chem Comm, 2016, 52:819-822) used a similar setup to prepare QDs from toluene (irradiation via Nd:YAG laser, 10 Hz repetition rate, 8 ns pulse width). Direct photodissociation of toluene molecules to carbon dots under pulsed laser irradiation in the absence of surfactants or catalysts was also reported by Zhu et al. (Carbon, 2016, 105:416-423). Authors should establish the novel aspects of the submitted manuscript in the Introduction section and clarify how their study advances the field.
Other comments:
FT-IR spectra in Fig 3e should also include pure toluene.
References are out of order throughout the manuscript. Reference 65 and 66 are the same paper.
Author Response
- My main concern is the novelty of synthesis employed in this study. Reference 66 (Yu et al., Preparation of carbon dots by non-focusing pulsed laser irradiation in toluene, Chem Comm, 2016, 52:819-822) used a similar setup to prepare QDs from toluene (irradiation via Nd:YAG laser, 10 Hz repetition rate, 8 ns pulse width). Direct photodissociation of toluene molecules to carbon dots under pulsed laser irradiation in the absence of surfactants or catalysts was also reported by Zhu et al. (Carbon, 2016, 105:416-423). Authors should establish the novel aspects of the submitted manuscript in the Introduction section and clarify how their study advances the field.
Our response: This study highlights the impact given by nanosecond laser radiation upon the size controlled of synthesis of GQDs from in an organic liquid ambience with exceptionally small reaction time. Apart from the previous report by Yu et al, here we explored the GQDs synthesis with second (532 nm) and third harmonic (355 nm) wavelengths other than the fundamental wavelength (1064 nm) without using any reaction protective gases like Argon. The whole experiment was done in room temperature and atmospheric pressure. We found that, at large fluences (F ≥ 3.1 Jcm−2), the ablation mechanism is apparently explosive boiling and does not lead to QDs formation and fabricated less than 10 nm GQDs at lowest laser fluences in this particular experimental conditions, apart from the study reported by Yu et al. Even though the direct photodissociation of toluene molecules to carbon dots under pulsed laser irradiation in the absence of surfactants or catalysts was also reported by Zhu et al (They used the Excimer Laser and the laser parameters are aslo different), we tried to generate GQDs in nanosecond regime and established the size control by varying the laser wavelength and established effect of the same to its NLO properties for the first time.
Other comments:
- FT-IR spectra in Fig 3e should also include pure toluene.
Our response: FT-IR spectra of pure toluene is displayed in Fig. 3f. Of note, after being laser irradiated, the FT-IR spectra of the GQDs suspension showed the same characteristics as pure toluene (compare spectra in fig. 1e and 1f), confirming that the solvent after laser irradiation was still toluene.
- References are out of order throughout the manuscript. Reference 65 and 66 are the same paper.
Our response: It is addressed and corrected in the revised version.
Round 2
Reviewer 2 Report
Authors addressed my concerns fully.